# Offline Next Generation Metagenomics Sequence Analysis Using MinION Detection Software (MINDS)

**DOI:** 10.3390/genes10080578

**Published:** 2019-07-30

**Authors:** Samir V. Deshpande, Timothy M. Reed, Raymond F. Sullivan, Lee J. Kerkhof, Keith M. Beigel, Mary M. Wade

**Affiliations:** 1Science and Technology Corporation, 111 Bata Blvd, Suite C, Belcamp, MD 21017, USA; 2US Army, 20th CBRNE, Aberdeen Proving Ground, MD 21010, USA; 3US Army, CCDC—Chemical Biological Center, Aberdeen Proving Ground, MD 21010, USA; 4Department of Marine and Coastal Sciences, Rutgers University, 71 Dudley Rd, New Brunswick, NJ 08901-8521, USA

**Keywords:** phylogenetic classification, visualization, third generation sequencing, offline analysis pipeline

## Abstract

Field laboratories interested in using the MinION often need the internet to perform sample analysis. Thus, the lack of internet connectivity in resource-limited or remote locations renders downstream analysis problematic, resulting in a lack of sample identification in the field. Due to this dependency, field samples are generally transported back to the lab for analysis where internet availability for downstream analysis is available. These logistics problems and the time lost in sample characterization and identification, pose a significant problem for field scientists. To address this limitation, we have developed a stand-alone data analysis packet using open source tools developed by the Nanopore community that does not depend on internet availability. Like Oxford Nanopore Technologies’ (ONT) cloud-based What’s In My Pot (WIMP) software, we developed the offline MinION Detection Software (MINDS) based on the Centrifuge classification engine for rapid species identification. Several online bioinformatics applications have been developed surrounding ONT’s framework for analysis of long reads. We have developed and evaluated an offline real time classification application pipeline using open source tools developed by the Nanopore community that does not depend on internet availability. Our application has been tested on ATCC’s 20 strain even mix whole cell (ATCC MSA-2002) sample. Using the Rapid Sequencing Kit (SQK-RAD004), we were able to identify all 20 organisms at species level. The analysis was performed in 15 min using a Dell Precision 7720 laptop. Our offline downstream bioinformatics application provides a cost-effective option as well as quick turn-around time when analyzing samples in the field, thus enabling researchers to fully utilize ONT’s MinION portability, ease-of-use, and identification capability in remote locations.

## 1. Introduction

Field-deployable instruments are quickly demonstrating the transition in rapid point-of-care diagnostics and bio-surveillance allowing for reliable detection and accurate therapeutic countermeasures [1,2,3]. Several companies have developed deployable technologies for molecular diagnostics and biodefense that perform fast sample-to-answer analysis in the field [4,5,6]. Use of this equipment in emergency response situations, such as an outbreak exposure to endemic infectious diseases or the intentional use of bioweapons, allows for rapid turnaround time and definitive results, which are critical to the health and security of the people within the community. Unfortunately, most of these technologies are restricted to pre-set assay panels that could miss pathogens outside their target reach and do not generally identify organisms with antimicrobial resistance and enhanced virulence. While these instruments are proven and reliable, the user is confined to the targeted panels, primers, probes or antibodies that can be carried in the field with them. The data output is also limited to either a small PCR amplicon or protein target, providing a very narrow sliver of the whole genomic picture.

Metagenomics and whole genome sequencing are increasingly being used for diagnostic and clinical laboratories for the detection of pathogenic organisms [7,8,9]. These features enable the lab to conduct genomic characterization and phylogenic analysis, which is critical towards understanding evolutionary change, virulence and transmission during an outbreak. Oxford Nanopore Technologies (ONT) has recently developed sequencing technology that allows the user to sequence virtually anywhere in the world and low Earth orbit [10,11,12]. This small, portable device also enables affordable bio-surveillance on a global scale [13,14]. The MinION device has been field tested with successful sequencing of arbovirus, Ebola virus and Zika virus [6,15,16,17,18,19]. For example, in 2018, Nigeria experienced a record upsurge in cases of Lassa fever. A multinational team under the auspices of the World Health Organization (WHO) partnering with the Nigeria Center for Disease Control used metagenomic data generated by the MinION to determine the outbreak was due to independent zoonotic transmission events and not a viral strain with an increased transmission rate. The research group was able to rapidly deploy field labs and obtain epidemiological information critical to understanding the spread of the epidemic [20]. 

However, one of the biggest challenges still facing MinION sequencing in the field is offline software needed to analyze the raw data. Ideally, this offline software will have a simple to use graphical user interface (GUI) that allows users without a strong understanding of command line code and computer science experience to perform analysis and determine actionable results. Unfortunately, current ONT downstream bioinformatics and characterization often requires an internet connection and/or coding experience, which generates a bottleneck in real-time analysis to most individuals. Even with connectivity to an institutional laboratory, delay can mean death in critical situations. 

One solution to this problem for next generation sequencing was the development of the Empowering the Development of Genomics Expertise (EDGE) including The Pan-Genomics for Infectious Agents (PanGIA) bioinformatics platform [21]. Sponsored by the Defense Threat Reduction Agency (DTRA), these platforms were designed to analyze Illumina short reads and were somewhat adapted for MinION long reads. In this paper, we demonstrate an offline downstream characterization pipeline specifically designed for MinION long reads. MINDS (MinION Detection Software) uses the open source read software Centrifuge for taxonomic classification purposes [22]. This real-time data streaming allows immediate analysis of the data, enabling rapid identification of bacteria, virus and fungi in a sample. Our MinION sequence analysis software provides offline real-time species identification and characterization on a standard laptop without the need for internet connectivity or high end computing power, thereby enabling true portability and validation of the samples in the field as well as in the lab.

## 2. Materials and Methods

### 2.1. Bacterial Sample

MSA-2002 was purchased from ATCC, Manassas, VA. The sample contains a mixture of 20 different bacterial strains distributed equally (5% ea.): *Acinetobacter baumannii* (ATCC 17978), *Actinomyces odontolyticus* (ATCC 17982), *Bacillus cereus* (ATCC 10987), *Bacteroides vulgatus* (ATCC 8482), *Bifidobacterium adolescentis* (ATCC 15703), *Clostridium beijerinckii* (ATCC 35702), *Cutibacterium acnes* (ATCC 11828), *Deinococcus radiodurans* (ATCC BAA-816), *Enterococcus faecalis* (ATCC 47077), *Escherichia coli* (ATCC 700926), *Helicobacter pylori* (ATCC 700392), *Lactobacillus gasseri* (ATCC 33323), *Neisseria meningitidis* (ATCC BAA-335), *Porphyromonas gingivalis* (ATCC 33277), *Pseudomonas aeruginosa* (ATCC 9027), *Rhodobacter sphaeroides* (ATCC 17029), *Staphylococcus aureus* (ATCC BAA-1556), *Staphylococcus epidermidis* (ATCC 12228), *Streptococcus agalactiae* (ATCC BAA-611), and *Streptococcus mutans* (ATCC 700610).

### 2.2. Metagenomic Sample Preparation

Working in a field-deployable laboratory requires the thorough vetting of equipment, processes and procedures prior to deployment due to the resource-limited environments normally encountered which would curtail sample preparation and analysis. The OmniLyse by Claremont BioSolutions provides several advantages when working in an austere environment. Effective lysis has been proven across a variety of cell types including gram-positive bacteria, spores, yeast, and cysts. Only 1–2 min is required to provide consistent yields of gDNA [23]. The small footprint of the OmniLyse and battery powered bead beating mechanism allows for easy use inside a glove box or biosafety cabinet when working with unknown, potentially high threat organisms. Fragmentation length can also be controlled based on the volume and lysis time of the sample. DNA cleanup was performed using Agencourt AMPure XP beads to provide high gDNA recovery and reduce the need for centrifugation. From start of extraction to final gDNA material, the total time is around 35 min. When using the Rapid Library Kit from Oxford Nanopore, the total time for library completion is one hour. Once loaded on the flow cell, the data collection can vary from a few thousand reads in an hour to over a hundred thousand reads in seven hours. Depending on the total number bases needed, the sample to answer using this method is from two to eight hours. 

Genomic DNA was extracted and purified from MSA-2002 (ATCC, Manassas, VA, USA) using OmniLyse (Claremont BioSolutions, Upland, CA, USA) mechanical disrupter for 5 min in phosphate buffered saline. Agencourt AMPure XP (Beckman Coulter, Brea, CA, USA) cleanup was performed with following modifications. A 0.5 sample volume of 5M NaCl (Fisher Scientific, Hampton, NH, USA) with 0.5 sample volume of 30% PEG, 1.5M NaCl (Fisher Scientific, Hampton, NH, USA) was added to the lysed cells. Then 50 μL of resuspended AMPure XP beads were added and allowed to bind for 15 min. The beads were washed two times with fresh solution of 70% ethanol (Fisher Scientific, Hampton, NH, USA). After removal of the ethanol, 10 μL of nuclease-free water (VWR, Radnor, PA, USA) was added and incubated at 55 °C for 10 min to elute the gDNA from the beads [24]. Library preparation was performed with Rapid Sequencing Kit SQK-RAD004 (Oxford Nanopore Technologies, Oxford, UK) following manufactures protocol. 400 ng of template DNA was incubated with fragmentation mix at 30 °C for 1 min and at 80 °C for 1 min and cooled at 4 °C. The tagmented genomic DNA was mixed with the rapid adapter mix for five minutes at room temperature. The prepared DNA library was placed on ice until loaded on the flow cell. Platform QC was run on an R9.4.1 revD MinION flow cell (Oxford Nanopore Technologies, Oxford, UK) prior to each sequencing run. 

### 2.3. Sample Sequencing and Bioinformatics Analysis

Reads acquisition (ONT’s *MinKNOW* core ver. 3.1.20 and base-calling (ONT’s Guppy software ver. 2.0.10) was integrated on ONT’s MinIT device (MinIT Release 19.01.10, 256 core GPU, 8 GB RAM, 512 GB SSD storage weighing 290 g) connected wirelessly to a Dell Precision 7720 laptop (Intel i7-7820HQ CPU, 4 core/8 thread, 2.9 GHz, 64 GB RAM with 3 TB SSD storage running Windows 10 Professional). The laptop was used primarily to run MINDS for downstream analysis (Figure 1). After 14 h of run time, FASTQ files were submitted to MINDS 1.0.53.

MINDS is a user-friendly GUI written in Microsoft *C#* incorporating Python (version 3.6) scripts for file handling, processing, reporting and read mapping using Centrifuge 1.0.4 to perform taxonomic classification. Report graphics were generated using matplotlib and Seaborn [25,26,27]. The reads were searched against an indexed database of all RefSeq bacterial and archeal genome sequences downloaded periodically from Centrifuge developer’s website [28]. MINDS is available from the corresponding author by request.

Low level sequence data noise/background (near neighbors, and false positives and negatives) was filtered by removing sporadic mapped reads to species—those having less than 0.1% of all total reads mapped and/or less than 5 unique mapped reads.

MINDS performance was compared with ONT’s cloud-based EPI2ME What’s In My Pot (WIMP) workflow [29,30], a Centrifuge based system and also with the taxonomic sequence classifier Kraken [31]. A standard Kraken database of all complete bacterial, archaeal and viral genomes in NCBI’s Reference Sequence (RefSeq) database was built on 28 June 2019 using a 72 core, 512 Gb RAM server located at Rutgers University, New Brunswick, NJ. The MinION sequence data was analyzed by Kraken on the same server. 

## 3. Results

Using ATCC MSA-2002 as a metagenomic mock community allowed us to test the MINDS pipeline with a variety of gram-positive and gram-negative bacteria. A 3 × 10^7^ cells equal ratio, whole cell mix containing 20 bacterial strains was lysed for five minutes using OmniLyse. After cleanup, the nucleic acid concentration was set to 53 ng/µL and library preparation was performed using SQK-RAD004. The metagenomic mock community sample was run overnight for 14 h. In the first eight hours 170,000 reads were generated. An additional six hours of run time acquired only 3440 more reads due to the unavailable pores in the flow cell. Nanoplot [32] was used to obtain the statistical results of this run (Table 1). Over 390 million bases were called and the mean Q score of 9.6. The basecalled FASTQ files generated were then submitted to Centrifuge and Figure 2 shows the results generated after filtering. The overall analysis from sample submission to report took 15 min. 

Nineteen of 20 species from the MSA-2002 mock community were correctly identified, with an additional 10 near-neighbor microorganisms also identified. Table 2 shows the *Centrifuge* results after filtering identifying 29 species in the sample listed by total and unique reads. Total reads are sequences classified to species level (including multi-classified reads). Unique reads are classified to a single species. *Actinomyces odontolyticus* was not identified because it was not present in the *Centrifuge* database, so reads were assigned to closely related *Actinomyces meyeri*. Interestingly, it was recently proposed that both species belong to *Schaalia*, a new genus in the *Actinobacteria* [33]. Comparing the genome sequences of *Schaalia odontolytica* (NCBI accession NZ_DS264586.1) and *Schaalia meyeri* (NCBI accession NZ_CP012072.1) results in a 79% sequence identity with a coverage of 74%. The low coverage is due to the genome size difference: 2.39 Mb for *Schaalia odontolytica* vs. 2.05 Mb for *Schaalia meyeri*. 

MINDS also identified nine additional bacterial species closely related to three MSA-2002 species. *Shigella dysenteriae*, *Shigella boydii*, *Shigella flexneri*, *Shigella sonnei* and *Shigella* sp. PAMC 28760 are close relatives of MSA-2002’s *E. coli* and belong to a pan-genomic group [34,35]. Likewise, *Bacillus thuringiensis*, *Bacillus anthracis* and *Bacillus* sp. ABP14 are close relatives and belong to the *Bacillus cereus* group [36] *Clostridium pasteurianum* is a close relative of *Clostridium beijerinckii* and has been mistaken for it recently [37]. 

Offline Centrifuge was compared with other read mappers to benchmark its accuracy. Table 3 categorizes the read mapper results from highest number of MSA-2002 reads to the lowest. ONT’s “What’s In My Pot” (WIMP) cloud-based classification pipeline was used as a baseline since this module also uses *Centrifuge* for identification. The FASTQ files were also compared with Kraken for analysis. The three read mappers produced similar results, except the following: Kraken mapped far fewer reads to *Streptococcus agalactiae*, but correctly mapped 778 reads to the recently re-classified *Schaalia odontolytica* (formerly *Actinomyces odontolyticus*) [33]. 

Strain level identification is an important goal of taxonomic classification. For example, it would be important for a commander to know whether *Bacillus anthracis* Ames or *Bacillus anthracis* Sterne was used in an attack: the former is deadly, the latter is a vaccine strain [38]. Table 4 shows the percentage of reads mapped to strain by WIMP and Kraken. Few strains had >90% reads mapped to them. The top strain hit (highest number of strain reads mapped per species) was correct for WIMP in 13 of 20 cases and for Kraken in 9 of 20 cases. 

## 4. Discussion

As sequencing continues to move into the field, great effort is needed to ensure the user has the necessary equipment and software required for detection. The OmniLyse kit along with Solid Phase Reversible Immobilization (SPRI) clean up provides a small consumable footprint for DNA extraction, removing the need of centrifuge and spin columns used in traditional extraction kits. Small, portable thermocyclers also allow the library preparation with ONT’s Rapid Sequencing Kit (RAD004) performed with little space requirements. This rapid extraction and purification method does have a tradeoff as the quality of the DNA is considerably lower than the values recommended by ONT, thus affecting the throughput of DNA. However, the quantity obtained using 188 ng/μL afforded sufficient gDNA for sequencing and would allow for possible refueling of the flow cell to increase the amount of data generated. 

Four repeated experiments were performed to increase the read output of the gram-positive *Actinobacteria*: *Cutibacterium acnes, Bifidobacterium adolescentis,* and *Actinomyces meyeri* (data not shown). However, these organisms were consistently underrepresented with respect to the total reads generated. Even with five minutes of OmniLyse cell disruption, no change in read distribution was observed. An extraction method with higher quality gDNA output for possible refueling the flow cell might be required if more genome coverage of these organisms is necessary. 

The recent release of ONT’s GPU-based MinIT greatly reduces the computational burden on the portable laptop and also allows for real-time basecalling with the ability to perform 150 k bases per second verses a traditional CPU-based computer with an output of 20 k bases per second. The user also has the ease of “plug and play” feature of the MinIT and not have to worry about the laptop’s capability with the MinION. Using MinIT for real time basecalling allowed us to have the FASTQ files ready for downstream analysis as soon as the sample acquisition on MinION was stopped. Future efforts will focus on customizing the MINDS pipeline to classify reads in real-time as the FASTQ files are generated from the MinIT. This has the potential to reduce the run time and more rapidly determine results allowing for faster decision and countermeasures in the case of a biothreat detection. 

Software in the field not only has to work offline, it also needs intuitive interface features that allows the end user unfamiliar with command line code the ability to quickly analyze data. The MINDS application provides easy to use graphical interface that minimizes the need for command line expertise (Figure 3). The end user simply submits the folder holding FAST5/FASTQ files along with other prevalent information such as flow cell ID, MinION serial number, etc. Once all the relevant information is submitted, the analysis can be performed by clicking the “Start” button. Based on the workflow selection, data analysis is performed, and a taxonomy report is generated. 

The MINDS pipeline has proven its successful adaptability during ONT’s release of new software and products. Since MINDS acts like a wrapper for the latest tools developed by the ONT community, it can be easily modified to accommodate the new software and kits released by ONT. For example, Guppy replaced Albacore and MINDS seamlessly integrated the new base-caller. With changes to the Centrifuge code, MINDS was able to analyze ONT’s change from single FAST5/FASTQ to multi FAST5/FASTQ. Lastly, as demultiplexing tools evolved through the past few years, MINDS implemented various open source software changes with no change to the GUI. The rapidly changing software development for processing MinION data requires regular patching or updating the command line code that interfaces with a stable GUI, which has proven easily accomplished with MINDS over the past few years. 

Centrifuge was chosen for its mapping utility on a laptop: it has a relatively small indexed database size and RAM requirement. For example, our indexed Kraken database was 227 Gb compared to the 25 Gb Centrifuge database, and RAM requirement for Centrifuge were ~4 Gb: approximately 20× less than Kraken’s. As shown, there is little discernable difference in the performance of the two read mappers. As more useful read mappers are developed, they can be easily substituted and incorporated into the MINDS interface. 

## 5. Conclusions

MINDS software was developed for users without a scientific education or laboratory background, such as enlisted soldiers, sailors, airmen and marines. Real-world metagenomic data can be difficult to analyze and interpret, especially for a user that is unfamiliar with bioinformatic tools. In contrast, MINDS allows any user to run sample data and receive quick, actionable results with a clear interpretation. For example, MINDS generates easy to understand bar graphs and pie charts, while providing the raw read information in a very intuitive graphical form. In this report, we have demonstrated an unbiased fieldable detection capability using ONT’s MinION sequencing platform and the MINDS platform. Our system dramatically reduces the time frame needed to detect targets as well as providing a sequencing in the field capability which minimizes the burden of overseas shipping of samples back to a lab such as the Centers for Disease Control or U.S. Army Medical Research Institute for Infectious Disease. Furthermore, the intuitive GUI of MINDS allows any user to quickly perform classification on their reads generated from MinIT. Additionally, simple parameter selection allows the user to provide percent cutoff to remove background noise to minimize false-positives and false-negatives which can interfere with the identification and decision-making processes. 

Several open-source software tools for classification were tested for field applications. Centrifuge performed faster than the other tools tested on the same computational hardware and did not require a large computational memory burden due to its database indexing capabilities. A small footprint database is a decisive feature for field forward computation. The offline Centrifuge classified all 20 organisms very similarly to the cloud-based Centrifuge through WIMP and also with Kraken. However, neither Centrifuge nor Kraken could convincingly classify the taxa to strain.

Future efforts will include having MINDS run data streaming from the MinION and MinIT in real time, enhancing MINDS capabilities to provide faster interpretation of the results. Also, further development in the sample preparation workflow is needed as library preparation still requires several hands-on steps with various pieces of laboratory equipment and consumables to operate in a field-forward environment. Efforts have begun to minimize this laboratory equipment burden to allow sequencing anywhere by anyone. These include future products developed by Oxford Nanopore including VolTRAX and Ubiq. Strain level identification is an important goal we hope to achieve by first assembling the reads into larger contigs before classification. Assembled contigs will provide a much smaller number, yet much longer sequences to map against the database and should provide more information rich strain determining features than the individual unassembled read sequences alone. 

## Figures and Tables

**Figure 1 genes-10-00578-f001:**
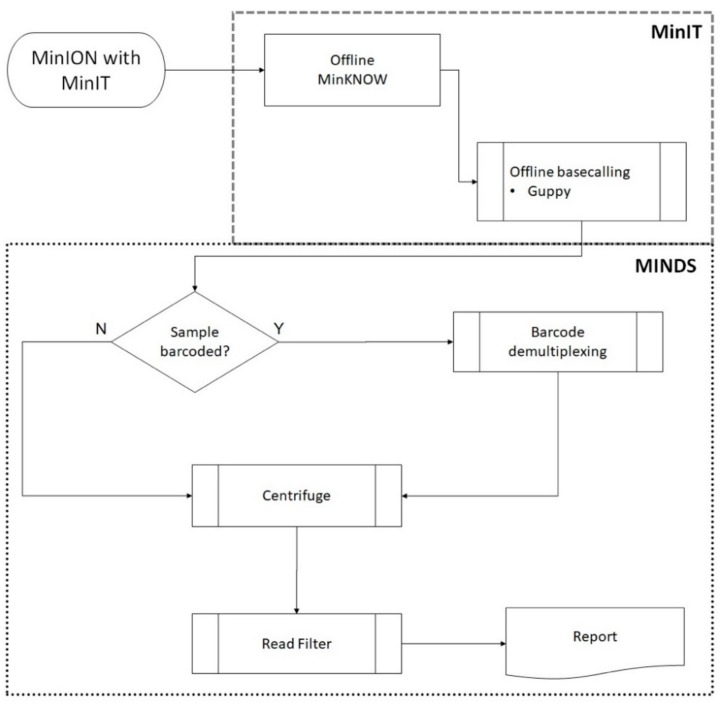
Schematic demonstrating sequencing and bioinformatic workflow. Reads from the MinION were acquired with Oxford Nanopore Technologies’ (ONT’s) MinIT real-time base-calling device running MinKNOW and Guppy software. The downstream MINDS workflow processed the metagenomic sequence data read wirelessly from the MinIT to identify microorganisms present in the sample. Barcoded samples were first demultiplexed (if required) using ONT’s qcat software. Centrifuge (Johns Hopkins Center Computational Biology) mapped reads to taxonomic classifications. Background noise was filtered by removing species with only sporadic reads: having less than 0.1% of all total reads mapped and/or having less than 5 unique reads mapped.

**Figure 2 genes-10-00578-f002:**
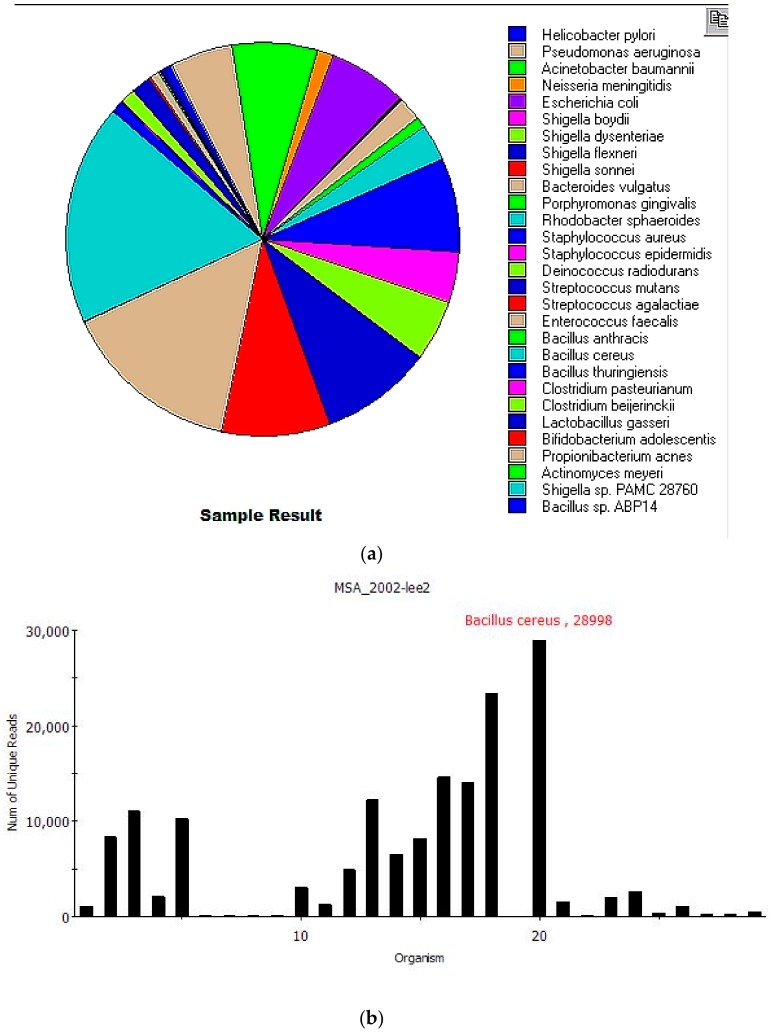
MINDS report showing 19 of the 20 MSA-2002 mock species correctly identified from with an additional 10 near neighbor species identified. (**a**) The pie chart displays unique read abundance of each species. *Actinomyces odontolyticus* was not present in the *Centrifuge* database, so reads were assigned to closely related *Actinomyces meyeri*. MINDS also identified nine additional bacterial species closely related to three MSA-2002 species: five *Shigella* spp. from the *E. coli/Shigella* pan-genomic group, three *Bacillus* spp. from the pan-genomic *Bacillus cereus* group and *Clostridium pasteurianum*, a near-neighbor of *Clostridium beijerinckii*. (**b**) The MINDS taxonomy report displaying a read abundance histogram of the 29 species identified.

**Figure 3 genes-10-00578-f003:**
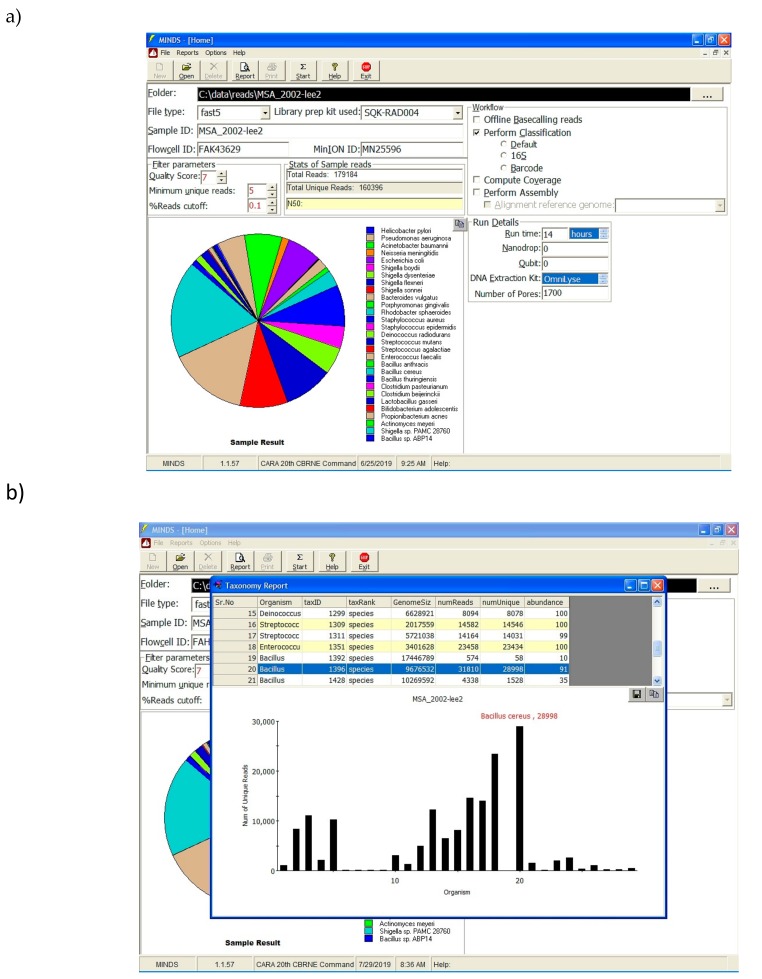
(**a**) MINDS graphical interface allows users to straightforwardly submit the FASTQ file, input experimental metadata and conditions, generates intuitive data analysis and (**b**) easily interpretable reports.

**Table 1 genes-10-00578-t001:** Nanoplot statistics for the MinIT base-calling output and post MinION Detection Software (MINDS) statistics for the Centrifuge analysis. The average multi-classified read mapped to 2.53 species (Centrifuge multi-classified reads count/Actual MinION reads that Centrifuge multi-classified).

Mean Read Length	2268 bp
Mean read quality	9.6
Medium read length	1344 bb
Median read quality	9.7
Number of reads	173,440 bp
Read length N50	4275 bp
Total bases	393,502,204 bp
Reads above quality cutoffs > Q5	173,440 (100%)
Reads above quality cutoffs > Q7	173,440 (100%)
Reads above quality cutoffs > Q10	68,766 (39.6%)
Reads above quality cutoffs > Q12	1344 (0.8%)
Reads above quality cutoffs > Q12	0 (0%)
Number of reads mapped by Centrifuge	184,795
Unclassified reads by Centrifuge	5611
Centrifuge total reads count	179,184
Centrifuge unique reads count	160,396
Centrifuge multi-classified reads count	18,788
Actual MinION reads that Centrifuge multi-classified	7433

**Table 2 genes-10-00578-t002:** Filtered *Centrifuge* results showing all 29 species identified in the MSA-2002 sample. Total reads are reads classified to species including multi-classified reads. Unique reads are those classified to a single species. The confidence score is the percentage of unique reads to total reads. The confidence grades are simply ranges of confidence scores: A = 90–100%, B = 80–90%, C = 70–80%, D = 60–70%, F = 0–60%. The last column shows the relative abundance of unique reads—the percentage of species specific unique reads to total unique reads. Except for the incorrectly identified *Actinomyces meyeri*, it can be seen that the other 19 MSA-2002 species had high unique reads to total reads ratios (confidence grades of B or better), while the incorrectly identified near-neighbors had low confidence grades.

Species	Total Reads	Unique Reads	Confidence Score	Confidence Grade	Relative Unique Reads
*Bacillus cereus*	31,810	28,998	91.16%	A	18.24%
*Enterococcus faecalis*	23,458	23,434	99.90%	A	14.74%
*Streptococcus mutans*	14,582	14,546	99.75%	A	9.15%
*Streptococcus agalactiae*	14,164	14,031	99.06%	A	8.82%
*Staphylococcus aureus*	13,056	12,270	93.98%	A	7.72%
*Acinetobacter baumannii*	11,289	11,113	98.44%	A	6.99%
*Escherichia coli*	12,479	10,248	82.12%	B	6.45%
*Pseudomonas aeruginosa*	8475	8403	99.15%	A	5.28%
*Deinococcus radiodurans*	8094	8078	99.80%	A	5.08%
*Staphylococcus epidermidis*	7118	6455	90.69%	A	4.06%
*Rhodobacter sphaeroides*	4907	4900	99.86%	A	3.08%
*Bacteroides vulgatus*	3266	3103	95.01%	A	1.95%
*Lactobacillus gasseri*	2584	2564	99.23%	A	1.61%
*Neisseria meningitidis*	2212	2113	95.52%	A	1.33%
*Clostridium beijerinckii*	2151	2011	93.49%	A	1.26%
*Porphyromonas gingivalis*	1282	1273	99.30%	A	0.80%
*Cutibacterium acnes*	1105	1093	98.91%	A	0.69%
*Helicobacter pylori*	1020	1012	99.22%	A	0.64%
*Bifidobacterium adolescentis*	353	341	96.60%	A	0.21%
*Actinomyces meyeri*	227	219	96.48%	A	0.14%
*Bacillus thuringiensis*	4338	1528	35.22%	F	0.96%
*Bacillus* sp. ABP14	1968	433	22.00%	F	0.27%
*Shigella* sp. PAMC 28760	1776	231	13.01%	F	0.15%
*Clostridium pasteurianum*	290	153	52.76%	F	0.10%
*Shigella boydii*	687	131	19.07%	F	0.08%
*Shigella dysenteriae*	376	98	26.06%	F	0.06%
*Shigella sonnei*	676	83	12.28%	F	0.05%
*Shigella flexneri*	580	78	13.45%	F	0.05%
*Bacillus anthracis*	574	58	10.10%	F	0.04%
Total	174,897	158,998	N/A	N/A	100%

**Table 3 genes-10-00578-t003:** Comparison of MINDS’ offline *Centrifuge* read mappers with ONT’s cloud-based *Centrifuge* (WIMP-What’s In My Pot) and *Kraken* read mappers. All three read mappers produced similar results except for *Streptococcus agalactiae* and *Actinomyces meyeri* as noted below.

Species	Offline Centrifuge	Online Centrifuge (WIMP)	Kraken
*Bacillus cereus*	28,998 (18.1%)	32,074 (19.2%)	28,205 (17.0%)
*Enterococcus faecalis*	23,434 (14.6%)	23,914 (14.3%)	23,656 (14.2%)
*Streptococcus mutans*	14,546 (9.1%)	14,902 (8.9%)	14,208 (8.6%)
*Streptococcus agalactiae*	14,031 (8.8%)	14,263 (8.5%)	4139 (2.5%)
*Staphylococcus aureus*	12,270 (7.7%)	12,685 (7.6%)	12,145 (7.3%)
*Acinetobacter baumannii*	11,113 (6.9%)	10,410 (6.2%)	8998 (5.4%)
*Escherichia coli*	10,248 (6.4%)	7411 (4.4%)	7226 (4.3%)
*Pseudomonas aeruginosa*	8403 (5.2%)	8600 (5.2%)	8327 (5.0%)
*Deinococcus radiodurans*	8078 (5.0%)	8314 (5.0%)	8090 (4.9%)
*Staphylococcus epidermidis*	6455 (4.0%)	6265 (3.8%)	6146 (3.7%)
*Rhodobacter sphaeroides*	4900 (3.1%)	5040 (3.0%)	4967 (3.0%)
*Bacteroides vulgatus*	3103 (1.9%)	3190 (1.9%)	3101 (1.9%)
*Lactobacillus gasseri*	2564 (1.6%)	2601 (1.6%)	2469 (1.5%)
*Neisseria meningitidis*	2113 (1.3%)	2163 (1.3%)	1965 (1.2%)
*Clostridium beijerinckii*	2011 (1.3%)	2308 (1.4%)	1965 (1.2%)
*Porphyromonas gingivalis*	1273 (0.8%)	1312 (0.8%)	1276 (0.8%)
*Cutibacterium acnes*	1093 (0.7%)	1114 (0.7%)	1090 (0.7%)
*Helicobacter pylori*	1012 (0.6%)	1091 (0.7%)	1048 (0.6%)
*Bifidobacterium adolescentis*	341 (0.2%)	354 (0.2%)	241 (0.2%)
*Actinomyces meyeri*	219 (0.1%)	229 (0.1%)	778 * (0.5%)
Other identified organisms	4191 (2.6%)	8706 (5.2%)	25,991 (15.6%)
**Total Reads**	**16,0396**	**16,6946**	**166,131**

* reads classified to *Schaalia odontolytica* (formerly *Actinomyces odontolyticus*).

**Table 4 genes-10-00578-t004:** Percentage of ATCC MSA-2002 reads mapped to strain by ONT’s cloud-based “What’s In My Pot” (WIMP) and offline Kraken taxonomic classifier. Strains are listed as they are referred to in NCBI’s RefSeq genome database. Reads and percentages in **boldface** had the highest number of strain reads mapped per species.

Strain	Online *Centrifuge* (WIMP)			*Kraken*		
	Species	Strain	% Strain Mapped	Species	Strain	% Strain Mapped
*Bacillus cereus* ATCC 10987	32,074	**24,115**	**75.2%**	28,205	**18155**	**64.4%**
*Enterococcus faecalis* OG1RF	23,914	**13,420**	**56.1%**	23,656	109	0.5%
*Streptococcus mutans* UA159	14,902	122	0.8%	14,208	42	0.3%
*Streptococcus agalactiae* 2603V/R	14,263	**1139**	**8.0%**	4139	242	5.8%
*Staphylococcus aureus* subsp. *aureus* USA300_FPR3757	12,685	223	1.8%	12,145	5	0.0%
*Acinetobacter baumannii* ATCC 17978	10,410	0	0.0%	8998	**637**	**7.1%**
*Pseudomonas aeruginosa* ATCC 9027	8600	**7507**	**87.3%**	8327	0	0.0%
*Deinococcus radiodurans* R1	8314	**8206**	**98.7%**	8090	**8090**	**100.0%**
*Escherichia coli* str. K-12	7411	31	0.4%	7226	170	2.4%
*Staphylococcus epidermidis* ATCC 12228	6265	**2506**	**40.0%**	6146	**1637**	**26.6%**
*Rhodobacter sphaeroides* ATCC 17029	5040	**4504**	**89.4%**	4967	**3646**	**73.4%**
*Bacteroides vulgatus* ATCC 8482	3190	**3190**	**100.0%**	3101	**2273**	**73.3%**
*Lactobacillus gasseri* ATCC 33323 = JCM 1131	2601	**2275**	**87.5%**	2469	**2076**	**84.1%**
*Clostridium beijerinckii* ATCC 35702	2308	3	0.1%	1965	1	0.1%
*Neisseria meningitidis* MC58	2163	**414**	**19.1%**	1965	6	0.3%
*Porphyromonas gingivalis* ATCC 33277	1312	**39**	**3.0%**	1276	9	0.7%
*Propionibacterium acnes* subsp. *defendens* ATCC 11828	1114	**388**	**34.8%**	1090	**176**	**16.1%**
*Helicobacter pylori* 26695	1091	1	0.1%	1048	3	0.3%
*Bifidobacterium adolescentis* ATCC 15703	354	**291**	**82.2%**	341	**228**	**66.9%**
*Actinomyces meyeri*	229	N/A	0.0%	778 *	N/A	0.0%

* reads classified to *Schaalia odontolytica* (formerly *Actinomyces odontolyticus*).

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
