# Peer review of "Offline Next Generation Metagenomics Sequence Analysis Using MinION Detection Software (MINDS)"

_genes, 2019, doi:10.3390/genes10080578_

Round 1
Reviewer 1 Report
I was a reviewer on the prior submission of this paper and my comments reflect that.
The paper is generally well written and the topic is important. This will be a very useful and practical tool for field-based sequencing, when the downstream analysis is basic, e.g. what microbial community members are present. The lack of an easy to use graphical user interface for such analyses is a problem.
In this revised version, the authors do compare performance to Centrifuge and Kraken although did not compare performance to Mash (Ondov et al. 2016). I would suggest a bit more discussion (like a few sentences) regarding the advantages/disadvantages of these approaches in the discussion. Is the proposed strategy of assembly + Centrifuge likely to result in strain-level identification?
I would have liked to hear more about the implementation. In the prior version, there was more description of this, e.g. implementation in Python. I would prefer to see more detail, not less.
Is MINDS available online? If not, why not? I did not see any mention of availability.
Page 3, line 98, end of sentence needs a period.
Previously, the number of reads acquired was 41,445. Now it says 170,000 in first 8 hours. Can you explain the discrepancy? I appreciate you including the nanoplot statistics.
Prior comment not addressed: Did you remove adaptors prior to analysis (PoreChop for example)? If not, consider doing this and assessing whether adaptor removal affects your results or not.
Most or all other questions have been addressed and the paper is an important contribution. The future work (real time, incorporating sample prep workflow advancements) will be very nice.
Author Response
Thank you for the comments and questions given for our manuscript. We greatly appreciate your time and giving your expertise to our paper. The following is a point-by-point response.
In this revised version, the authors do compare performance to Centrifuge and Kraken although did not compare performance to Mash (Ondov et al. 2016). I would suggest a bit more discussion (like a few sentences) regarding the advantages/disadvantages of these approaches in the discussion. Is the proposed strategy of assembly + Centrifuge likely to result in strain-level identification?
Thanks for the comment. Please note in the discussion, the last paragraph was added discussing read mapper performance.
We were unaware of MASH, but have downloaded it and run it against our reads. It looks very interesting. However, we quickly realized its use will require more time than the Genes editors will allow us for this manuscript. We will look at MASH for our follow-on project.
A sentence was added to the end of the conclusions explaining why we think assembly+Centrifuge will likely result in stain level identification.
I would have liked to hear more about the implementation. In the prior version, there was more description of this, e.g. implementation in Python. I would prefer to see more detail, not less.
Thank you very much for this, we overlooked this point. We added a short description of what MINDS is, a GUI written in C# with python ver 3.6 used for file handling and processing in section 2.3.
Is MINDS available online? If not, why not? I did not see any mention of availability.
The corresponding author (Wade) will provide the software upon request and a signed user agreement required by the US Army.
Page 3, line 98, end of sentence needs a period.
Done.
Previously, the number of reads acquired was 41,445. Now it says 170,000 in first 8 hours. Can you explain the discrepancy? I appreciate you including the nanoplot statistics.
In the time between your first and second review, we performed more runs to try to increase the read output of the gram-positive Actinobacteria: Cutibacterium acnes, Bifidobacterium adolescentis, and Actinomyces meyeri. The fourth run is the experiment cited.
Prior comment not addressed: Did you remove adaptors prior to analysis (PoreChop for example)? If not, consider doing this and assessing whether adaptor removal affects your results or not.
In order to streamline Minion signal acquisition and basecalling, we used the MinIT device (Release 19.01.10 ) which integrated Guppy ver. 2.0.10. The Guppy version did not integrate adapter removal ability. Subsequent Guppy versions do, but upgrading Guppy on MinIT has proven challenging for ONT and we are still waiting for it. Adapter removal would slightly improve the average Q score from 9.6 to 9.7 and remove 268 “bad” reads. We did not think the small improvement warranted incorporating Porechop in MINDS since the next Guppy version on the MinIT will obviate the need.
We've also attached the word document for our response.

Reviewer 2 Report
In the revised manuscript changes in
the responses for my comments have been prepared in a clear and
sufficient way.
Author Response
Thank you for your time and expertise regarding our manuscript.
This manuscript is a resubmission of an earlier submission. The following is a list of the peer review reports and author responses from that submission.
Round 1
Reviewer 1 Report
The authors develop an "offline" MinION metagenomic analysis pipeline based primarily on the use of Centrifuge. They test their approach on a sample dataset and achieve good accuracy.
I have two major issues:
1) Comparison with other methods
The authors have done a good job in testing their approach on a real metagenome library based on 20 different organisms. However, a useful addition to the study would be to compare the accuracy of their results to that obtained by other MinION analysis pipelines - even ones that are "online". This would help with interpretation of their pipeline's results.
Since the sequencing data has already been generated, this should be quite easy to do. Comparing their results to at least 2 other methods would benefit the study.
2) The Figures are presented poorly and several revisions need to be made.
* Figure 3 - text is much too small
* Figs 2 and 3 - the colour palette in the pie charts of Figure 2 and 3 are inconsistent; this should definitely be fixed to be made consistent.
* All text/font sizes in Figures must be made the same.
* Figure legends - all legends need more description of the content of the figures. E.g., The Figure 4 legend is only "WIH Report".
* Figure 4 - why is there a heatmap shown when the actual values are also in the table. This figure should be removed and converted to a table instead since the figure adds no value.
Reviewer 2 Report
The Authors attempt to develop offline downstream bioinformatics applications for identification of microorganisms without the need for internet connectivity. Field-deployable equipment is invaluable in an emergency response situation. However, many devices of this kind have certain limitations. In my opinion, the application of this sequence analysis software is a promising area from the point of view of microbial community analysis. For this reason, this work can be considered as essential and valuable.
Abstract
Line 30: Please insert space between (1) and Background.
Introduction
Line 52: Please, insert space between fever and [21].
Line 54: Please, check the bracket with WHO.
Materials and Methods
Line 78: Please add a subsection in the entire Methodology chapter, e.g. 2.1. Bacterial sample instead ‘Sample information’. It will organize the methodology more.
Line 79: The word ‘Table” should be written with a capital letter. Please, check the spaces.
Line 94: Please, insert space between 75 and µL.
Results
Figure 2-3: In my opinion, the figures are illegible and should be corrected. Moreover, please remove the title ‘Sample Result’ from figure’s area. Figure 3 and 4 show the same results. One of them should be removed.
There is no explanation why Propionibacterium acnes appears in both raw and generated after WIH results.
Discussion
The chapter ‘Discussion’ is a results summary rather than their discussion. The Authors should discuss obtained results in the light of previous studies or the use of such software/devices. Potential future application areas may also be highlighted and discussed.
Lines 218-220: Are you sure this fragment did not appear in the manuscript by mistake?
Reviewer 3 Report
The paper is generally well written and the topic is important. This will be a very useful and practical tool for field-based sequencing, when the downstream analysis is basic, e.g. what microbial community members are present. The lack of an easy to use graphical user interface for such analyses is a problem.
The authors do mention the importance of a graphical user interface, but do not really describe the prior solutions for offline analysis of (nanopore) sequencing data, such as MASH (Ondov et al. Genome Biology 2016 17:132), Kraken (Wood and Salzberg Genome Biology 2014 15:R46), KrakenUniq (Breitwieser et al. Genome Biology 2018 19:198). Notably, the authors do mention Centrifuge. By describing these tools, authors can highlight some of the challenges, for example, MASH is not highly sensitive and specific, but is very fast, and lightweight in terms of computation and memory. Kraken is very sensitive, but requires a large database, with commensurate requirements for RAM. This requirement makes Kraken challenging for use in the field, where typical computers may not have enough RAM. It would also be good to compare your solution (Centrifuge under the hood) to the sensitivities and specificities achieved using other solutions, such as MASH and Kraken.
Detailed points:
page 2, line 52, insert space before ref [21]
Table 1: As all composition percentages are the same, it seems reasonable to omit this column and include this as a footnote?
page 3, line 103: please explain what p_compressed means and what is included in the p_compressed database. All refseq genomes? A selected subset?
page 3, line 105: please explain the relevance of scikit-learn… e.g. the underlying code was written in Python and utilized the scikit-learn package to implement linear regression (just a little more detail regarding implementation might be useful)
Figure 1 is very clear
Extraction protocol is very clear
Choice of OmniLyse is good as it has previously been demonstrated to achieve highly quantitative extraction results (Irwin et al. BMC Microbiol 2014 Dec 31; 14:326).
Page 4, line 135: 41445 seems like a relatively small number of reads for 11 hours… can you provide some additional statistics? How many bases? How long were reads on average?
Did you remove adaptors prior to analysis (PoreChop for example)? If not, consider doing this and assessing whether adaptor removal affects your results or not
Page 4, line 135: “uploaded” – by uploaded do you mean transferred from MitIT to computer? If so, I would use another word as “uploaded” may suggest internet connection to some readers.
Page 4, line 136: How did you chose the 3% threshold?
Page 4, line 142: insert “the” before “WIH”
Page 5, line 147: “within 15 minutes” – was this 15 minutes since start of sequencing, and analysis was real time? Or is this 15 minutes after starting analysis, after 6 hours of sequencing? Please clarify in the text if the analysis was real-time or batch, after completion of sequencing. Real-time would be preferable for many applications. If not real time, could you make your system real-time by monitoring read files as they become available on MinIT?
Page 5, line 150: “distribution of reads” sounds vague to me… I suggest using different language such as “percentage of reads.”
Page 5, line 152: similar as prior comment regarding use of word “distribution”
Page 5, line 155: suggest insert the word “the” before “NT database”
Page 5, Figure 2 is fuzzy; can you provide a higher resolution image?
Page 5, Figure 3 text is not legible. Can you enlarge the text to make it legible?
Page 5, is it possible to use an alternative chart mechanism to show how reads originally binned in one taxonomy end up in another after WIH analysis?
Page 6, line 178: “150 bases per second” vs. “20 bases per second” – this seems wrong. A MinIT could be expected to handle ~5G bases over 48 hours, or 29k Bases/s. Can you please re-examine your estimates?
Page 6, lines 187-198 and Figure 5: Did you use the GUI in your analysis? If so, I would suggest moving this to methods, so the process for using the software is part of the experiment you describe. If not, that should be made more clear that the GUI implements the underlying process but wasn’t used for the experiment.
Page 7, line 224: you mention the “sample to answer time” – what was this for your experiment? If this was reported, I missed it.
Page 7, line 224: can you elaborate further how your tool contributes to simplifying sample logistics (and what this means)?